# How to Study the Location and Size of Rectal Tumors That Are Candidates for Local Surgery: Rigid Rectoscopy, Magnetic Resonance, Endorectal Ultrasound or Colonoscopy? An Interobservational Study

**DOI:** 10.3390/diagnostics14030315

**Published:** 2024-01-31

**Authors:** Anna Serracant, Beatriz Consola, Eva Ballesteros, Marta Sola, Francesc Novell, Noemi Montes, Xavier Serra-Aracil

**Affiliations:** 1Coloproctology Unit, General and Digestive Surgery Department, Institut d’Investigació i Innovació Parc Tauli I3PT-CERCA, Parc Tauli Hospital Universitari, Universitat Autònoma de Barcelona, Parc Tauli s/n, 08208 Sabadell, Spain; annserracant@gmail.com (A.S.); montes.ortega@gmail.com (N.M.); 2Diagnostic Radiology Department, Institut d’Investigació i Innovació Parc Tauli I3PT-CERCA, Parc Tauli Hospital Universitari, Universitat Autònoma de Barcelona, Parc Tauli s/n, 08208 Sabadell, Spain; bconsola@tauli.cat (B.C.); eballesteros@tauli.cat (E.B.); martasolagarcia@gmail.com (M.S.); fcescn@gmail.com (F.N.)

**Keywords:** rectal tumors, rectal MRI, local surgery

## Abstract

1. Background. Preoperative staging of rectal lesions for transanal endoscopic surgery (TES) comprises digital rectal examination, intraoperative rigid rectoscopy (IRR), endorectal ultrasound (EUS), colonoscopy and rectal magnetic resonance imaging (rMRI). The gold standard for topographic features is IRR. Are the results of the other tests sufficiently reliable to eliminate the need for IRR? rMRI is a key test in advanced rectal cancer and is not operator-dependent. Description of anatomical landmarks is variable. Can we rely on the information regarding topographic features provided by all radiologists? 2. Materials and Methods. This is a concordance interobservational study involving four diagnostic tests of anatomical characteristics of rectal lesions (colonoscopy, EUS, rectal MRI and IRR), performed by four expert radiologists, regarding topographic rectal features with rMRI. 3. Results. Fifty-five rectal tumors were operated on by using TES. The distance of the tumor from the anal verge, location by quadrants, size by quadrants and size of tumor were assessed (IRR as gold standard). For most of the tumors, the correlation between IRR and colonoscopy or EUS was very good (ICC > 0.75); the correlation between rMRI and IRR in respect of the size by quadrants (ICC = 0.092) and location by quadrants (ICC = 0.292) was weak. Topographic landmarks studied by the expert radiologists had an excellent correlation, except for distance from the peritoneal reflection to the anal verge (ICC = 0.606). 4. Conclusions. Anatomical description of rectal lesions by IRR, EUS, colonoscopy and rMRI is reliable. Topographic data obtained by EUS and colonoscopy can serve as a reference to avoid IRR. Determination of these topographic data by rMRI is less reliable. As performed by the expert radiologists, the anatomical study by rMRI is accurate and reproducible.

## 1. Introduction

Rectal cancer is a complex condition, accounting for one-third of colorectal cancers with non-negligible morbidity and mortality [1,2]. The initial study of a rectal lesion (RL) is the basis for defining the treatment. Depending on the diagnostic TNM, there are several treatment options: neoadjuvant treatment, and surgical and adjuvant approaches.

In early locoregional stages without metastases, the initial study of RL is key to determining the best surgical strategy. Options range from local surgery (transanal endoscopic surgery (TES)) to total mesorectal excision (TME). Moreover, it determines the need for neoadjuvant treatment or a combination of neoadjuvant treatment with TES (i.e., T2-sT3 without pathological nodes) [3]. A colonoscopy provides topographic information about RL (location, height, appearance) and obtains biopsies for studying the pathology. Endorectal ultrasound (EUS) and rectal magnetic resonance imaging (rMRI) are the diagnostic tests used for tumor staging (T) and lymph node involvement (N) [4,5]. Each test provides relevant information, but their conclusions are not definitive.

EUS is associated with rigid rectoscopy, which allows an assessment of the height and location of an RL. EUS evaluates T (63–95% accuracy) and N (63–85% accuracy) [1]. Its main drawback is that it is operator-dependent [2,6]. rMRI provides topographic data and evaluates T and N [7]. It also provides information about the peritoneal reflection and radial margin [8].

Anatomical references used for surgery differ from those used by radiologists to interpret rMRI. The interpretation of these findings is complex and requires experience; there may be disagreement even between expert radiologists [9]. Furthermore, several anatomical guidelines are used to analyze the images obtained by rMRI, without any clear consensus on the one that is most suitable for this purpose [10].

TES is indicated in adenomas and T1 adenocarcinomas [11] whose height is at their maximum at 18–20 cm from the anal verge (AV) [12]. When performing TES, precise information is needed on the height of the RL in relation to the AV; it is known that RLs located 5 cm above or below the AV have a different pathological response [13]. The RL’s relation to the peritoneal reflection must also be assessed, due to the risk of perforation of peritoneal cavity [14]. The location by quadrants is also important for the positioning of the patient on the operating room table.

To perform TES, the exact topographic characteristics of the RL are required. Before surgery, an intraoperative rigid rectoscopy (IRR) is performed [12], with the patient in a supine position. Once topographic characteristics are confirmed, the patient’s position may be changed.

The main objective of this study is to assess the concordance between IRR and colonoscopy, EUS and MRI regarding RL topographic characteristics. The secondary objectives are related to answering the following questions: Is rMRI reliable enough to avoid systematic performance of IRR? Is rMRI more reliable than colonoscopy and EUS for determining the topographic RL characteristics? Are there differences in the interpretations of rMRI made by expert radiologists regarding RL topographic characteristics?

## 2. Materials and Methods

### 2.1. Study Design

An observational study was performed of the interobserver agreement regarding patients undergoing TES with curative intent consecutively between four diagnostic tests: colonoscopy, EUS, rMRI and IRR. To determine the variability between radiologists’ interpretations, an interobservational agreement study was carried out by four radiologists who are experts in rMRI.

This study followed the Ethical Principles for Medical Research Involving Human Subjects as outlined in the Declaration of Helsinki. Approval was obtained from the Clinical Research Ethics Committee (CEIm) of Parc Taulí University Hospital (CEIm Parc Tauli) (ID: 2017592, Approval Date: 13 July 2017). The STROBE guidelines for observational studies were followed [15]. Informed consent was obtained from the patients after an explanation of the risks and benefits of the procedure.

Data were recorded prospectively and analyzed retrospectively. Computerized data management was carried out using Microsoft^®^ Access 2003, introduced in a protected format.

### 2.2. Patients and Settings

This study was carried out at Parc Taulí University Hospital (Sabadell). It included fifty-five patients who had undergone consecutive TES with curative intent at the colorectal unit.

All RLs were studied according to the preoperative study protocol from our center [12]. In this process, RL candidates for TES (after preoperative FCS, EUS and rMRI tests) are classified depending on preoperative surgical indication: group I, curative intent (preoperative biopsy of adenoma), for patients who, after EUS (us) and rMRI (mr), are staged as us-mr T0–T1 and us-mr N0; group II, curative intent (preoperative biopsy of low-grade adenocarcinomas), us-mr T0–T1 and us-mr N0; group III, consensus indication (low-grade adenocarcinomas), us-mr T2 and us-mr N0, for patients who reject radical surgery; group IV, palliative indication; and group V, atypical indications [16].

Colonoscopies were performed by gastroenterologists. EUS and rigid rectoscopy were performed by surgeons from our colorectal unit. rMRIs were assessed by expert radiologists. The definition of an expert radiologist is one that has a minimum of 5 years’ experience informing this test with a minimum of 20 cases per year.

When TES was indicated, IRR was performed by the surgeon prior to insertion of TEM/TEO (Figure 1). All the variables related to the tumor were checked (size, morphology, location by quadrants, height relative to the AV of the proximal and distal edge of the lesion). IRR is considered the gold standard for the topographic study of RLs [8,12].

### 2.3. Inclusion Criteria

Patients with preoperative indication of curative TES (groups I–II); lesions less than 15 cm from the anal verge.

### 2.4. Exclusion Criteria

Patients in preoperative indication groups III, IV and V; patients who, after intraoperative evaluation for TES, underwent abdominal surgery because TES was technically impossible; patients who, due to technical or patient-specific circumstances, did not undergo any of the three tests before this study. 

**Figure 1 diagnostics-14-00315-f001:**
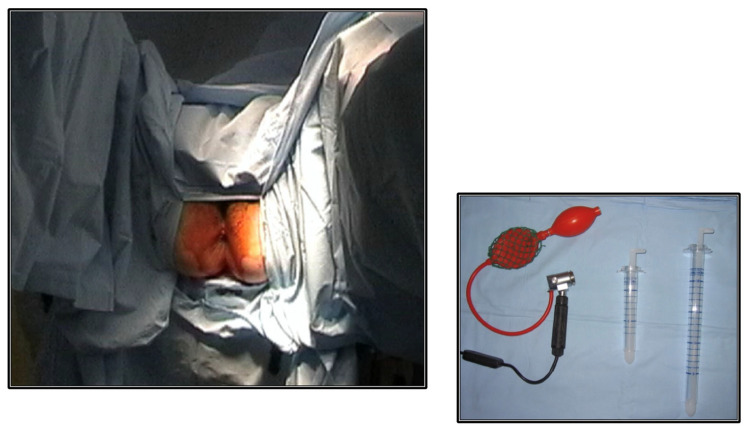
Before surgery, an intraoperative rigid rectoscopy is performed, with the patient in a supine position.

### 2.5. Preoperative Preparation, Surgical Technique

All patients with an indication of TES underwent mechanical colon preparation with antibiotic and thromboembolic prophylaxis [12]. Anesthesia was mainly general, except when the anesthesiologist recommended spinal anesthesia. The techniques used for TES were either TEM (Richard Wolf, Knittlingen, Germany) or transanal endoscopic operation (TEO, Karl Storz GmbH, Tüttlingen, Germany) [17].

### 2.6. Study Variables 

Interobservational study of the four tests, colonoscopy, EUS, rMRI, IRR:

Topographic variables of the tumor: height of the lesion with respect to the AV; quadrant occupied by the lesion; lesion size; lesion size by quadrants.

Interobservational study involving four expert radiologists in rectal MRI:

Distance from the tumor to mesorectal fascia; distance from the peritoneal reflection to the AV; distance from the lower margin of the tumor to the AV; distance from the upper margin of the lesion to the AV; distance from the upper margin of the tumor to the peritoneal reflection; distance from the lower margin of the lesion to the peritoneal reflection; distance to the radial margin.

### 2.7. Statistical Analysis

The sample size was initially considered to be twice the estimate of a large sample size (30 patients), resulting in the inclusion of 60 patients. From these consecutive patients, five of them were excluded based on the exclusion criteria, leading to the evaluation of 55 patients. A large sample was needed to ensure that the concordance analysis obtained results with the highest possible validity.

Statistical analysis was performed by using SPSS version 21 program (SPSS, Inc., Chicago, IL, USA).

In the interobserver agreement analysis for categorical dichotomous variables, Cohen’s Kappa index (KI) was used, which adjusts the effect of chance on the proportion of the observed agreement. For the interobserver agreement analysis of the continuous variables, Fisher’s intraclass correlation coefficient (ICC) was applied. The ICC allows evaluation of the general agreement between two or more measurement or observation methods based on an analysis of variance (ANOVA) model with repeated measures.

The Kappa Index and Fisher’s ICC were qualitatively interpreted as poor–weak (values less than 0.40), moderate (0.41–0.60), good (0.61–0.80), and very good (values greater than 1).

Two analyses were carried out: 1. an interobserver analysis between the gold standard (IRR) and each of the diagnostic tests (colonoscopy, EUS and rMRI) using KI, and the analysis between all of them using ICC; 2. an interobserver analysis of the items studied in rMRI according to the four expert radiologists using ICC.

## 3. Results

Of the 55 consecutive patients selected for TES with curative intent, 1 was excluded (indication was made following endoscopic polypectomy; pathological exam indicated adenocarcinoma with involvement of a resection margin). No lesion was identified by EUS or MRI; a minimal scar was observed, which was removed by TES. No lesion was found in the definitive pathological analysis. Therefore, the total sample studied comprised 54 patients [18].

Table 1 shows the demographic characteristics, features of the tumor identified by IRR, and pathology results of the lesions. The results are characteristic of patients with an indication of TES with curative intent: predominance of males (34, 63%), median tumor size 4 cm (IQR 50) (range: 1–9), location by quadrants mainly posterior (23, 42.6%) and mainly adenomas in pathology exam (70.4%).

Table 2 shows the results of the concordance using the KI between the IRR and the other approaches (rMRI, EUS and colonoscopy) and the concordance using Fisher’s ICC of the four tests for the same measure. 

The KI was very good in the distance from the lower margin of the tumor to the AV between IRR and rMRI, EUS and colonoscopy. Likewise, the ICC in all the tests was excellent (0.969, 95% CI: 0.948–0.982).

For the location of the lesion by quadrants, the KI between IRR and rMRI obtained a weak correlation. The KI between IRR and EUS was good, and between IRR and colonoscopy, it was very good.

For the evaluation of the size of the lesion by quadrants, the KI between IRR and rMRI presented a weak correlation. The KI between IRR and EUS was very good, and moderate between IRR and colonoscopy. The ICC for the correlation between all tests was not performed for either of the items.

Finally, in assessing the size of the lesion, the KI between IRR and rMRI, and colonoscopy was good. The KI between IRR and EUS was very good. The ICC of all the tests for this item was excellent.

Table 3 shows the results of calculating Fisher’s ICC to evaluate the correlation between four expert radiologists for the analysis of an rMRI. In total, 50 rMRIs were analyzed. In most of the variables (distance from the tumor to mesorectal fascia, distance from the lower margin of the tumor to the AV, distance from the upper margin of the tumor to the AV, distance from the upper margin of the tumor to the peritoneal reflection, distance from the lower margin of the tumor to the peritoneal reflection, distance to the radial margin or mesorectal fascia), excellent correlations were observed, except in the case of the distance of the peritoneal reflection to the AV, where it was merely good.

## 4. Discussion

The study of an RL lesion includes digital rectal examination, rigid rectoscopy, EUS, colonoscopy, rMRI and thoracoabdominal CT [2,4]. After the biopsy, and once the benign or malignant nature of the lesion is known, it is important to determine its height with respect to the anal verge, its size and its location according to the quadrant. In this setting, rigid rectoscopy is considered the gold standard, performed in the operating room. The present study aims to assess the reliability of the topographic findings obtained with three other techniques (EUS, colonoscopy and rMRI).

Colonoscopy is the examination of choice for the diagnosis of colon and RL. Quality indicators of a colonoscopy are complete examination, time between colonoscopies, size of the resected lesions and degree of colon cleansing [19]. However, the measurement of the height of the lesion in relation to the AV, especially in the rectum, sigma and left colon, is not always accurate; frequently, colonoscopy reports of a lesion’s height are not borne out by the results of the rigid rectoscopy. From the point of view of surgical strategy, this is extremely important.

It is very important to be accurate in the determination of the height of the lesion in the rectum. Perhaps unexpectedly, our study showed very good agreement between colonoscopy and IRR for determining the height of the lesion, the quadrant it occupies and its size. The agreement for size by quadrants was moderate. These results show that the test has good reliability compared to the gold standard.

In this study, we found a very good correlation between rMRI and IRR for defining the height and size of the lesion but not for determining the size by quadrants or its location by quadrants. Therefore, IRR could not be avoided when using MRI alone, so the quadrant occupied by RL cannot be well characterized, and it is important when planning TES.

EUS is operator-dependent, although with experienced raters, the reliability of the test increases. EUS has good precision for the diagnosis of tumors in initial stages, since it can differentiate between the distinct layers of the rectum, though its assessment of N is less accurate [1,5]. EUS can determine the size of the lesion, the quadrant it occupies and, therefore, the size by quadrants. There is no consensus on its ability to determine the height of the lesion, but a rigid rectoscopy is performed before the introduction of the rectal ultrasound [1,20,21]. Our study confirms a very good correlation in all the topographic items.

Characterization of an RL by rMRI is complex. The best way to describe these lesions and their anatomical relationships is by following a pre-established schema [8,22]. It is important to determine the distance of the tumor from the AV (upper, middle or lower margin); its distance from the circumferential margin, as its proximity to the mesorectal fascia implies a worse prognosis [23,24]; location of the tumor with regard to the peritoneal reflection; presence or absence of extramural venous involvement; and accurate assessment of sphincter apparatus and pelvic floor muscles.

Regarding T staging, one of the limitations of rMRI is its poor ability to differentiate between a T1 (confined to submucosa) and a T2 (confined to muscularis propia) due to its difficulty in differentiating between both layers [19,20,25]. It can correctly define whether the tumor is a superficial or deep T3, and whether it affects the visceral peritoneum or progresses beyond the rectum (T4). Similarly, rMRI is a good tool for the study and detection of N.

Discrepancies between radiologists’ interpretations of rectal MRI are well known [8]. In our study, a secondary objective was to assess the variability regarding expert radiologists’ impressions of the topographic characteristics in rMRI. We found that the agreement between our four radiologists on the anatomy of the lesion and its distance from the AV, the peritoneal reflection and the radial margin was excellent, and it was good for the distance from the peritoneal reflection to the AV. It is very important to determine the affection of the peritoneal reflection. Its affection could eliminate the need for neoadjuvant treatment. The correlation between radiologists is considered good, so the results are reliable.

Regarding the issue of whether rMRI findings provide enough information to make IRR unnecessary, we conclude that the judgment of an expert radiologist in characterizing an RL is reliable, but that, even so, IRR cannot be avoided.

Our results did not suggest that rMRI is more reliable than colonoscopy and EUS for assessing the topographic characteristics of the tumor. Evaluating the distance of the tumor from the anal verge and its size suggests that it may be equivalent (the ICC between MRI, FCS, EUS and IRR was excellent). But rMRI had a weak KI in comparison with IRR as it was not as accurate as IRR in determining the location and size by quadrants.

The findings of this study suggest that IRR could be avoided by using the joint results of EUS, FCS and MRI. Before each EUS, a rigid rectoscopy is already performed and the agreement between the two exams is very high. If the results are discordant, IRR is necessary.

## 5. Limitations of this Study/Future Directions

Some centers do not have access to EUS. rMRI is the only diagnostic test available to study local characteristics of RLs. Given the limitations of rMRI for differentiating between a T1 and a T2, and, sometimes, between a T2 and a superficial T3 [22], the use of EUS can further define the diagnosis of the tumor stage. Agreement between expert radiologists at our center was good or excellent. We now plan to carry out an interobservational study to assess whether our radiologists are able to differentiate between a T1 and a T2 by means of rMRI, comparing the results with EUS.

## 6. Conclusions

The anatomy of RLs that are candidates for TES can be reliably assessed by IRR, EUS, colonoscopy and rMRI. Topographic data obtained by EUS (combined with rigid rectoscopy) and colonoscopy can serve as a reference to avoid IRR, but this is not so in the case of rMRI. However, considering the results of the three tests together, IRR could be avoided. 

On the other hand, the expert radiologists’ assessment of the anatomy of RL by rMRI is reliable.

## Figures and Tables

**Table 1 diagnostics-14-00315-t001:** Patients’ demographic and preoperative variables (in surgery).

Variables			Results (*n* = 54 Patients)
Demographic Preoperative	Age (median–IQR–range) years		66 (IQR 17) (range: 42–87)
	Sex	Male	34 (63%)
		Female	20 (37%)
Tumor-related (in surgery)	Distance from distal margin to anal verge (median–IQR–range) cm		7 (IQR: 4.3) (range: 3–15)
	Distance from proximal margin to anal verge (median–IQR–range)		10 (IQR: 5) (range: 4.5–16)
	Size (medianIQR–range)		4 (IQR 50) (range: 1–9)
	Size by quadrants		
	Lesion morphology	Flat	18 (33.4%)
		Polypoid	16 (29.6%)
		Sessile	17 (31.5%)
		Ulcerated	6 (5.6%)
	Location by quadrant	Anterior	15 (27.8)
		Left lateral	10 (18.5%)
		Right lateral	6 (11.1%)
		Posterior	23 (42.6%)
	Indication for surgery	I	38 (70.4%)
		II	9 (16.7)
		III	7 (13%)
Pathology	Adenomas		39 (72%)
	Adenocarcinomas	T1	11 (20.4%)
		T2	2 (3.7%)
		T3ab	2 (3.7%)

**Table 2 diagnostics-14-00315-t002:** Results of concordance between the IRR and rMRI, EUS and colonoscopy.

	Cohen’s Kappa Index (95% CI)/Interpretation			Interclass Correlation Coefficient (95% ci)/Interpretation
Intraoperative Rectoscopy (IRR)	MRI	EUS	Colonoscopy	
	0.870 (0.757–0.931) Very good	0.981 (0.968–0.989) Very good	0.872 (0.770–0.928) Very good	0.969 (0.948–0.982) Excellent
Location by quadrants	0.292 (0.023–0.396) Weak	0.746 (0.655–0.881) Good	0.913 (0.788–0.996) Very good	-
Size by quadrants	0.092 (0.005–0.174) Weak	0.815 (0.722–0.913) Very good	0.439 (0.333–0.594) Moderate	-
Lesion size	0.758 (0.516–0.88) Good	0.805 (0.627–0.898) Very good	0.660 (0.276–0.839) Good	0.922 (0.869–0.957) Excellent

**Table 3 diagnostics-14-00315-t003:** Results of calculating Fisher’s ICC to evaluate the correlation between the four expert radiologists for the analysis of an rMRI.

Measurement Evaluated	ICC (Interclass Correlation Coefficient)	95% Confidence Interval	Interpretation
Distance from tumor to mesorectal fascia	0.817	0.539–0.948	Excellent
Distance from peritoneal reflection to anal verge	0.606	0.341–0.786	Good
Distance from lower margin of tumor to anal verge	0.969	0.948–0.982	Excellent
Distance from upper margin of tumor to anal verge	0.958	0.93–0.976	Excellent
Distance from upper margin of tumor to peritoneal reflection	0.724	0.498–0.863	Excellent
Distance from lower margin of tumor to peritoneal reflection	0.918	0.852–0.959	Excellent
Distance to radial margin or mesorectal fascia	0.836	0.565–0.958	Excellent

Fisher’s interclass correlation coefficient (ICC).

## Data Availability

Data is contained within the article.

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
