# Peer review of "How to Study the Location and Size of Rectal Tumors That Are Candidates for Local Surgery: Rigid Rectoscopy, Magnetic Resonance, Endorectal Ultrasound or Colonoscopy? An Interobservational Study"

_diagnostics, 2024, doi:10.3390/diagnostics14030315_

Round 1
Reviewer 1 Report
Comments and Suggestions for Authors Long-term results are missing, respectively the condition of the patients 3 and 5 years postoperatively. There is no information about adjuvant or neo-adjuvant treatment. The statistical study methods used in the article should be specified more clearly.Author Response
Long-term results are missing, respectively the condition of the patients 3 and 5 years postoperatively.
In this study, we are not presenting the results of the follow-up of the patients. Instead, we are discussing the utility of four tests in the diagnosis of a rectal lesion that is considered for local surgery.
There is no information about adjuvant or neo-adjuvant treatment.
No information in this field is discussed because we are focusing on rectal lesions considered for local surgery. These lesions include benign adenomas or malignant T1 adenocarcinomas, which are not considered for this treatment
The statistical study methods used in the article should be specified more clearly.
For the description of the quantitative variables, the mean and standard deviations were given. The categorical variables were described in absolute numbers and percentages.
In the interobserver agreement analysis for categorical dichotomous variables, Cohen's Kappa index (KI) was used, which adjusts the effect of chance on the proportion of the observed agreement. For the interobserver agreement analysis of the continuous variables, Fisher's intraclass correlation coefficient (ICC) was applied. The ICC allows evaluation of the general agreement between two or more measurement or observation methods based on an analysis of variance (ANOVA) model with repeated measures.
The Kappa index and Fisher's ICC were qualitatively interpreted as: poor-weak (values less than 0.40); moderate (0.41-0.60); good (0.61-0.80); and very good (values greater than 1).
Two analyses were carried out: 1. An interobserver analysis between the gold standard (IRR) and each of the diagnostic tests (colonoscopy, EUS and rMRI) using KI, and the analysis between all of them using ICC; 2. An interobserver analysis of the items studied in rMRI according to four expert radiologists using ICC.
Reviewer 2 Report
Comments and Suggestions for Authors
I read with interest the present study that covers an important issue of rectal lesions staging. The authors describe in details the correct assessment of the lesion and provide a detailed insight of the different diagnostic tools. I have the following comments:
- how was the sample size calculated? The authors report only the need for a ‘large number’
- the discussion should start with the main findings of the study
- the discussion could be implemented with what already present in literature on the comparison of the different diagnostic tools. Statistical reports could add value to the present research
Author Response
I read with interest the present study that covers an important issue of rectal lesions staging. The authors describe in details the correct assessment of the lesion and provide a detailed insight of the different diagnostic tools. I have the following comments:
- how was the sample size calculated? The authors report only the need for a ‘large number’.
The sample size was initially considered to be twice the estimate of a large sam-ple size (30 patients), resulting in the inclusion of 60 patients. From this consecutive patients, five of them were excluded based on exclusion criteria, leading to the eval-uation of 55 patients.

- the discussion should start with the main findings of the study.
We have modified the order of the paragraphs, positioning the paragraph with the main results of the study after the introduction of the discussion
"The study of a RL lesion includes digital rectal examination, rigid rectoscopy, EUS, colonoscopy, rMRI and thoracoabdominal CT [2,4]. After the biopsy, and once the benign or malignant nature of the lesion is known, it is important to determine its height with respect to the anal verge, its size, and its location according to the quadrant. In this setting, rigid rectoscopy is considered the gold standard, performed in the operating room. The present study aims to assess the reliability of the topographic findings obtained with the other techniques (EUS, colonoscopy, and rMRI).
Colonoscopy is the examination of choice for the diagnosis of colon and RL. Quality indicators of a colonoscopy are: complete examination; time between colonoscopies; size of the resected lesions, degree of colon cleansing [ ]. However, the measurement of the height of the lesion from AV, especially in rectum, sigma and left colon, is not always accurate; frequently, colonoscopy report of a lesion’s height is not borne out by the results of the rigid rectoscopy. From the point of view of surgical strategy, this is extremely important.
In the rectum, it is very important to be accurate in the determination of the height of the lesion. Perhaps unexpectedly, our study showed very good agreement between co-lonoscopy and IRR for determining the height of the lesion, the quadrant it occupies, and its size. The agreement for size by quadrants was moderate. These results show that the test has good reliability compared to the gold standard.
In this study we found a very good correlation between rMRI and IRR for defining the height and size of the lesion, but not to determine the size by quadrants nor its location by quadrants. Therefore, IRR could not be avoided using MRI alone, so the quadrant oc-cupied by RL cannot be well characterized, and it is important when planning TES."
- the discussion could be implemented with what already present in literature on the comparison of the different diagnostic tools. Statistical reports could add value to the present research.
We didn’t find other papers that discusses the comparison of the results of these four tests on the diagnosis of a rectal lesion between them. Neither one that compares the results of the study of a rectal lesion with RM between different radiologists. So, we are not able to compare it because it doesn’t’ exist.